# Effect of Heat Treatment Process on Microstructure and Mechanical Properties of High-Carbon H13 Steel

Kunda Du [1], Zhifeng Lv [2], Weichao Fan [2], Ruikong Zhang [2], Xuexian Li [2] and Lipeng Xu [1,*]

[1] School of Mechanical and Automotive Engineering, Liaocheng University, Liaocheng 252000, China; dkd1499285255@163.com

[2] Shandong EAST Engineering Tools Limited Liability Company, Liaocheng 252000, China; lzf-1020@163.com (Z.L.); 18963507870@163.com (W.F.); sdyst110419@126.com (R.Z.); q564818@163.com (X.L.)

* Correspondence: xulipeng@lcu.edu.cn

**Abstract:** This paper investigated the mechanical properties and microstructures of different samples of H13 steel after they underwent various heat treatment processes. It provided a detailed analysis of the microstructure and mechanical properties under different processes and approached the topic from a theoretical perspective. The phase composition of each sample remained unchanged after undergoing different heat treatment processes. Despite the vacuum gas quenching (H1) sample being guaranteed a hardness of 58.47 HRC, its toughness fell below expectations at a mere 46.75 J. Notably, the microstructure of the sample which underwent the H1 process and the cryogenic (H2) treatment exhibited a finer grain size and higher toughness compared to the sample which only underwent the H1 process without the cryogenic treatment. Its toughness was 70.19 J, but its hardness slightly decreased to 57.47 HRC. Following the application of oil quenching and cryogenic treatment (H3), the hardness of the sample significantly increased, reaching a remarkable 58.38 HRC. Additionally, the sample exhibited good impact resistance, with a measurement of 74.25 J. Before the H2 process, the sample which underwent the spheroidizing annealing process (H4) had a higher hardness compared to the sample without spheroidizing annealing. At the same time, when comparing the above four samples, the sample that underwent the H4 process exhibited the best toughness, with a value of 86.94 J, while still maintaining a hardness of 57.85 HRC; thus, it achieved an ideal balance between strength and toughness. Therefore, the optimal heat treatment process for high-carbon H13 steel was spheroidizing annealing followed by vacuum gas quenching and then cryogenic treatment.

**Keywords:** high-carbon H13 steel; heat treatment; hardness; impact resistance; mechanical property





## 1. Introduction

The cutter head is an essential and vital component in the construction of the hard rock roadheader. Additionally, the hob serves as a sharp and indispensable tool during the excavation process of the cutter head. The smooth advancement of the roadheader during the construction process relies entirely on the hob's performance. The selection of the hob cutter ring material and the formulation of the heat treatment process are two pivotal factors that have a decisive impact on the ultimate performance of the cutter ring. Therefore, it is of the utmost practical importance to conduct a comprehensive investigation and analysis of the hob cutter ring material and its corresponding heat treatment process.

The main failure modes of the hob cutter ring include wear, abnormal wear, cracking, curling, and so on. The failure of the cutter ring has a direct impact on both the construction period and the construction cost. The common domestic materials used for disc hob cutter rings include 9Cr2Mo, 6Cr4W2Mo2V, 40CrNiMo, 4Cr5MoSiV1 steel, and so on. These materials are selected based on their exceptional wear resistance or high toughness properties. Recently, shield tunneling has been evolving towards challenging construction scenarios involving large diameters, significant burial depths, and extensive distances.

Consequently, conventional materials struggle to meet the demands imposed by such intricate operational conditions. Therefore, the development of alloys with high hardness, wear resistance, and toughness is crucial in expanding the application scope and enhancing the service life of hob rings. 4Cr5MoSiV1 (H13), classified as a high-quality alloy steel, possesses high hardenability, excellent resistance to hot cracking, and good strength and toughness after medium- to high-temperature tempering. It finds widespread application in the manufacturing of hot-work molds and cutter rings [1]. However, for the exceptionally hard rock encountered by large shield machines, the rock-breaking cutter ring must possess not only extremely high hardness but also high toughness. However, H13 steel exhibits only average hardness and wear resistance due to its low carbon content.

In order to enhance the performance of hot-work die steel, scholars both domestically and internationally have conducted a series of research studies. Yin et al. [2] conducted a study on the effects of different austenitization and tempering temperatures on the structure and properties of 5Cr5Mo2V hot-forging die steel. The experiments revealed that the maximum quenching hardness was achieved at a temperature of 1030 °C. Additionally, a secondary hardening peak value was observed when tempering was performed at 550 °C. However, it was noted that the impact toughness was at its lowest during this tempering stage. Finally, it was concluded that the best heat treatment process is quenching at 1030 °C + tempering at 600 °C, which greatly improves the performance of 5Cr5Mo2V steel. Tarang et al. [3] studied the influence of cryogenic treatment on the corrosion behavior of H13 die steel. The research shows that the grain size of H13 die steel is obviously reduced and that the open circuit potential is increased after conventional heat treatment and then cryogenic treatment at −185 °C; this is beneficial to the improvement of the corrosion resistance of the material and can be used in an environment with high mechanical stress and corrosion. Melika [4] studied the effect of quenching and tempering treatment on the microstructure and wear resistance of H13 hot-work die steel. The experiment showed that the hardness of H13 hot-work die steel is greatly improved after austenitizing at 1050 °C and oil quenching, but slightly decreases after quenching at 530 °C in a single tempering treatment. A secondary hardening appears after quenching two times in 530 °C tempering treatment, and the hardness and wear resistance are greatly improved. Karkalos et al. [5] sorted out and summarized the hardening processing methods of various steel grades, and they introduced numerical simulation models and statistical methods and deeply summarized the application of these methods in the hardening process of steel, which could be used for reference by scholars. Peng et al. [6] added a high-temperature thermal refining process at 1100–1300 °C before the conventional heat treatment of H13 hot-work die steel. The results showed that under this process, the segregation structure was reduced, the overall structure uniformity was obviously improved, and the impact properties of the steel were greatly improved. Helen et al. [7] studied the rapid machining methods of H13 die steel and D2 tool steel; they studied the influence of different cutting tools and cutting media on the cutting effect, compared the effects of the two steels in the same processing environment, and proved the feasibility of various machining methods; this provided experienced guidance for the processing and application of H13 die steel and D2 tool steel. Wang et al. [8] studied the influence of quenching and tempering temperature on the microstructure and mechanical properties of H13 steel using a univariate analysis method. The results showed that the yield strength and austenite grain size increased with the increase in quenching temperature in the range of 1020–1060 °C. When tempering at 570~610 °C, with the increase in temperature, the impact toughness and elongation increase, and the hardness decreases. The heat treatment combination of quenching at 1040 °C and tempering at 570 °C achieves the best match between strength and toughness. The above research clarifies the influence of heat treatment temperature on hot-work die steel and gives a feasible scheme with which to improve the properties of hot-work die steel.

However, the effects of preparatory heat treatment, different quenching media, and the cryogenic process on hot-work die steel have not been clearly analyzed, and further discussion is needed. Building on this foundation, the study employs modified H13 as

the primary material to explore the influence of different quenching mediums on the microstructure and mechanical properties of H13 steel. Simultaneously, the mechanism underlying the effects of preheat treatment and cryogenic treatment on the modification of H13 steel is further elucidated. This will enable the optimization of the heat treatment process for H13 hot-work die steel, with the aim of enhancing the balance between hardness and toughness and ultimately achieving the optimal combination of material strength and toughness. It provides a reference for the application and promotion of improved H13 steel in the field of large shield tunneling.

## 2. Materials and Methods

The steel used in this test was forged modified H13 produced without heat treatment by a domestic special steel factory. Its main chemical composition is shown in Table 1:

**Table 1.** Chemical composition of experimental steel (mass fraction%).

| C | Si | Mn | Cr | Mo | V | Nb |
|---|---|---|---|---|---|---|
| 0.58 | 1.0 | 0.4 | 4.85 | 1.4 | 0.9 | 0.05 |

The room temperature impact specimens were selected according to their positions relative to the axis of the cutter ring; these positions were named outer axis, outer diameter, inner diameter, and inner axis. The samples were all processed using a wire cutting + grinding machine, and the unnotched standard specimens were prepared according to the ISO 148-1-2016 [9] carbon steel impact test standard. As shown in Figure 1, its size was 55 mm × 10 mm × 7 mm, and the impact test was carried out on a JB-300B pendulum testing machine produced by China Jinan Time Testing Instrument Co., Ltd. in Jinan, China. A Zeiss optical microscope manufactured by Carl Zeiss CMP GmbH in Jena, Germany and a scanning electron microscope were used to observe the microstructure after different heat treatments. The etchant used for the microstructure observation was 3% nitric acid alcohol and the test standard was implemented in accordance with ISO 945-1-2019 [10]. After the sample underwent grinding and polishing, the material phase was analyzed using a Brooke D8 advanced X-ray powder diffractometer manufactured by Brook AXS Co., Ltd., Germany, (Beijing, China), and the scanning angle range was 20 to 85, in accordance with the relevant standards of ISO 21068-1-2008 [11]. Using the HR-150A hardness tester produced by China Nanjing Wode Analytical Instrument Manufacturing Co., Ltd. in Nanjing, China, the samples were tested for Rockwell hardness. The test was conducted in accordance with ISO 6508-1-2016 [12]. For the different samples after heat treatment, the same surface was selected as the hardness test surface (size is 55 mm × 10 mm). Taking the geometric center of this surface as the symmetrical center, a rectangular area of 20 mm × 10 mm was selected as the hardness test area, as shown in Figure 2. In each sample, four points were randomly selected in the rectangular area as hardness test points, and the average and standard deviation of the four measured values were calculated. The wear test was carried out using a BD4603 wear tester produced by WenDeng AllWin Electric Machinery Co., Ltd. in Weihai, China. The test was carried out in accordance with the relevant standards of ISO 7148-1-2012 [13]. The diameter of the sample's wear surface was d = 15.5 mm. The Korean DEERFSO YA531+ abrasive belt, measuring 100 × 912 in size, with a mesh number of 80, was used. The test was performed at room temperature, with a loading mass of 255 g, a rotating speed of 2980 r/min, and a wear time of 40 min. The microstructure of the metal surface was observed using a double-beam FIB scanning electron microscope (FIB-SEM GX4) manufactured by Thermo Fisher Scientific Shier Technology in Waltham, MA, USA and the distribution of the interface elements was observed using its own EDS energy spectrum.

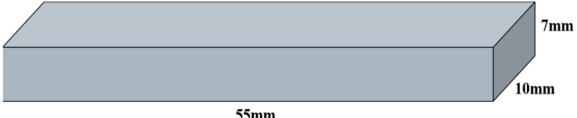

**Figure 1.** Structure diagram of impact test sample block.

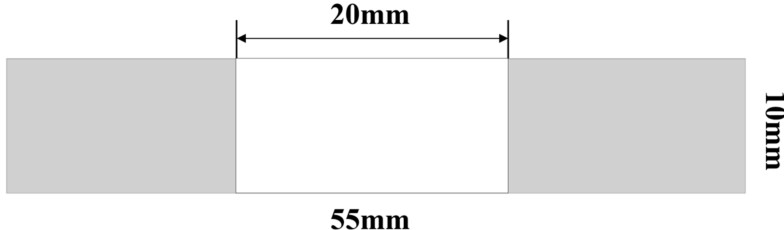

**Figure 2.** Hardness test area.

Sample heat treatment: the HNC2-1288 double-chamber vacuum quenching furnace was used for the oil quenching heat treatment. The gas quenching heat treatment was performed using the HZQ-150 horizontal high-pressure gas quenching vacuum furnace. The Cryometal-2000 L liquid nitrogen cryogenic box was employed to conduct the cryogenic treatment on the samples following the heat treatment. The same tempering process was applied to all the samples following quenching, using the following specific process: Tempering at 540 ± 10 °C for the first time, and keeping the temperature for 180 min; tempering at 540 ± 10 °C for the second time and keeping the temperature for 180 min; and tempering at 520 ± 10 °C for the third time and keeping the temperature for 180 min.

The specific heat treatment process of the sample was as follows:

(1) H1 (vacuum furnace gas quenching): ① Entered the furnace at room temperature, raised the temperature to 650 °C, and kept the temperature for 150 min; ② raised the temperature to 850 °C and kept the temperature for 120 min; ③ raised the temperature to 1040 °C and kept the temperature for 90 min; ④ vacuum gas quenched to a temperature below 80 °C for furnace discharge and left at room temperature, as shown in Figure 3a.

(2) H2 (vacuum gas quenching–deep cooling process): ① Entered the furnace at room temperature, raised the temperature to 650 °C, and kept the temperature for 150 min; ② raised the temperature to 850 °C and kept the temperature for 120 min; ③ raised the temperature to 1040 °C and kept the temperature for 90 min; ④ vacuum gas quenched to a temperature below 80 °C for furnace discharge; ⑤ deep cooling temperature of −80 °C, insulation for 90 min, cooled to around −30 °C and taken out of furnace and left at room temperature, as shown in Figure 3b.

(3) H3 (vacuum oil quenching–deep cooling process): ① Entered the furnace at room temperature, raised the temperature to 650 °C, and kept the temperature for 150 min; ② raised the temperature to 850 °C and kept the temperature for 120 min; ③ raised the temperature to 1040 °C and kept the temperature for 90 min; ④ vacuum oil quenched to a temperature below 40 °C for furnace discharge; ⑤ deep cooling temperature of −80 °C, insulation for 90 min, cooled to around −30 °C and taken out of furnace and left at room temperature, as shown in Figure 3c.

(4) H4 (spheroidizing annealing–vacuum gas quenching–deep cooling process): ① Heated to 860 ± 10 °C, kept the temperature for 120 min, quickly cooled to 700 ± 10 °C (near A1 line), and then kept the temperature for 480 min. Afterwards, cooled at a rate of 30–50 °C/h to 600 °C for furnace discharge, air cooled to room temperature, and completed spheroidization; ② entered the furnace to raise the temperature to 650 °C and kept the temperature for 150 min; ③ raised the temperature to 850 °C and kept the temperature for 120 min; ④ raised the temperature to 1040 °C and kept the temperature for 90 min; ⑤ vacuum gas quenched to a temperature below 80 °C for furnace

discharge; ⑥ deep cooling temperature of −80 °C, insulation for 90 min, cooled to around −30 °C and taken out of furnace and left at room temperature, as shown in Figure 3d.

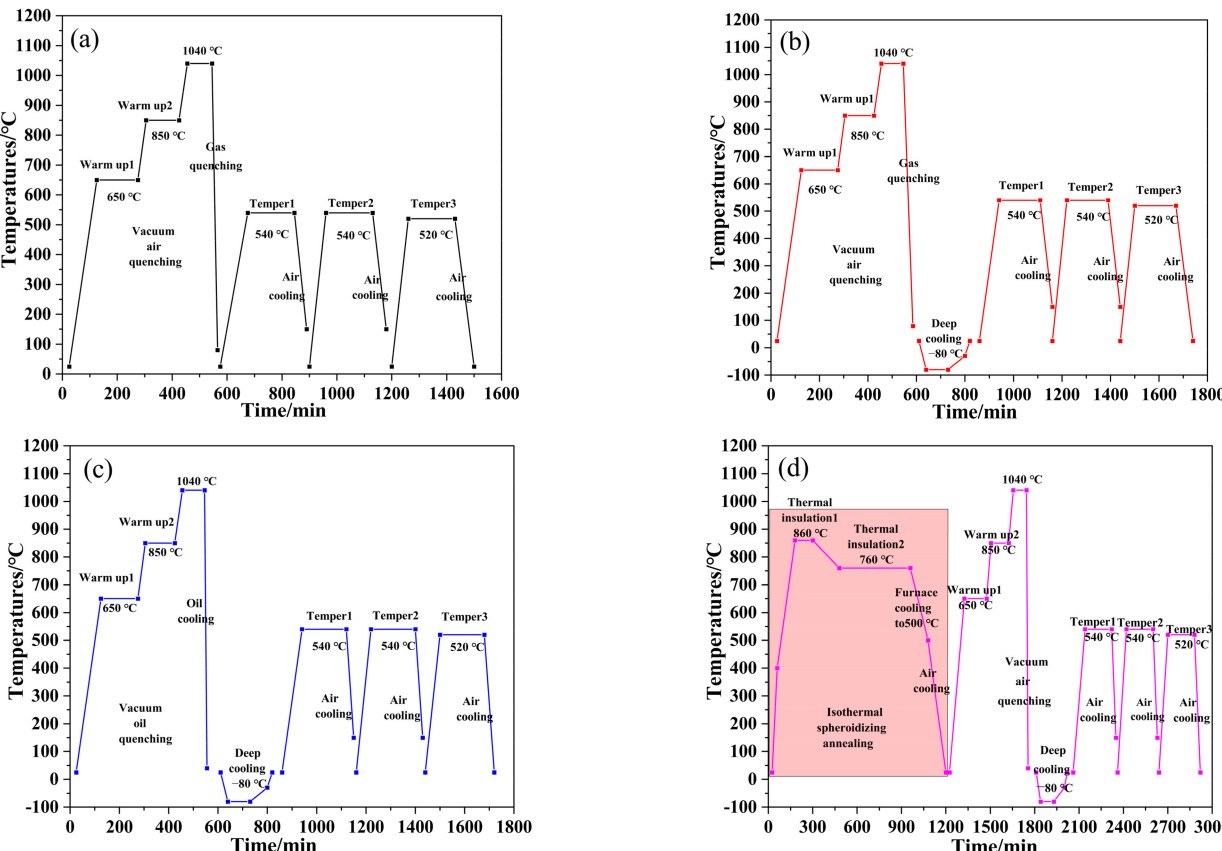

**Figure 3.** Schematic diagrams of heat treatment processes: (**a**) H1, (**b**) H2, (**c**) H3, (**d**) H4.

## 3. Experimental Results and Discussion

### 3.1. Effect of Heat Treatment Process on Microstructure

Figure 4 presents an EDS element area scan of H13 steel. As depicted in Figure 4a, the H13 steel samples exhibited granular substances of varying sizes. As observed in Figure 4b, the composition of the H13 steel in this particular area primarily consisted of the elements C, Cr, and Fe. The presence of bright spots representing the elements in Figure 4c,d, as well as the occurrence of dark spots in Figure 4e, provided visual evidence of the pronounced elemental segregation phenomenon in H13 steel. As can be seen from Figure 4d,e, the granular substances dispersed inside the structure were formed by C and Cr. As the Cr element is a strong carbide-forming element, it is easy to synthesize with carbon in steel to form chromium carbide, which plays a dispersion strengthening role and thereby improves the wear resistance of the H13 matrix.

In order to further explore the influence of the heat treatment methods on the structural elements of H13 steel, the samples after different heat treatments were analyzed by EDS. Figure 5a–d shows the EDS area scan analysis diagrams of the samples of H13 steel after the different heat treatments, respectively. Based on the evidence depicted in Figure 5, it is apparent that the element composition of H13 steel exhibited slight variations following the different heat treatment processes. It was evident that the heat treatment method exerted a minimal influence on the composition of H13 steel, and the slight discrepancies in each sample's composition might be attributed to equipment detection errors and the potential introduction of impurity elements during the heat treatment process [14].

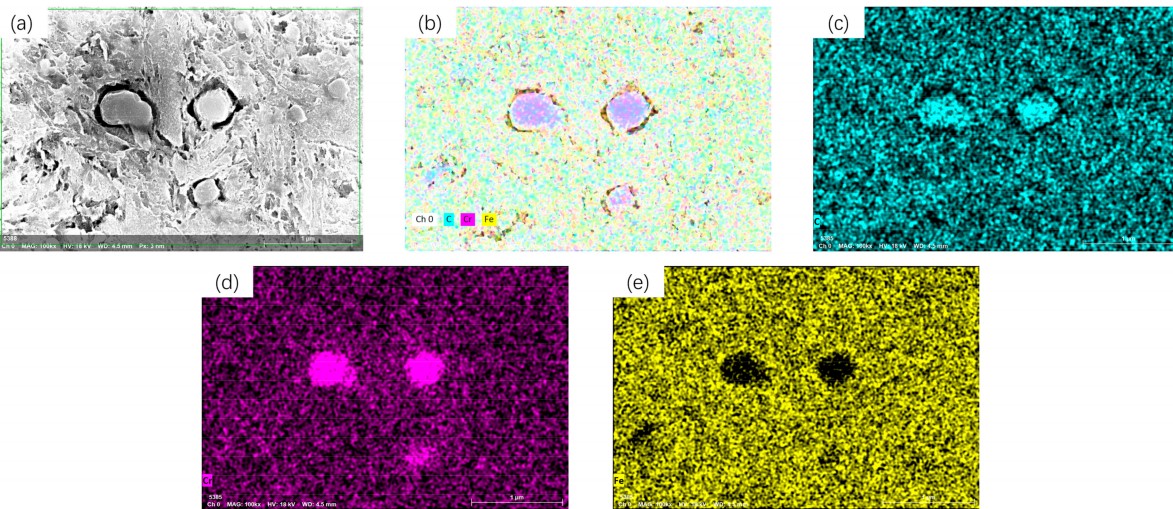

**Figure 4.** SEM image and EDS element distribution of H13 steel: (**a**) SEM photos of selected areas; (**b**) element distribution in selected area; (**c**) distribution of carbon element; (**d**) distribution of chromium element; (**e**) distribution of iron element.

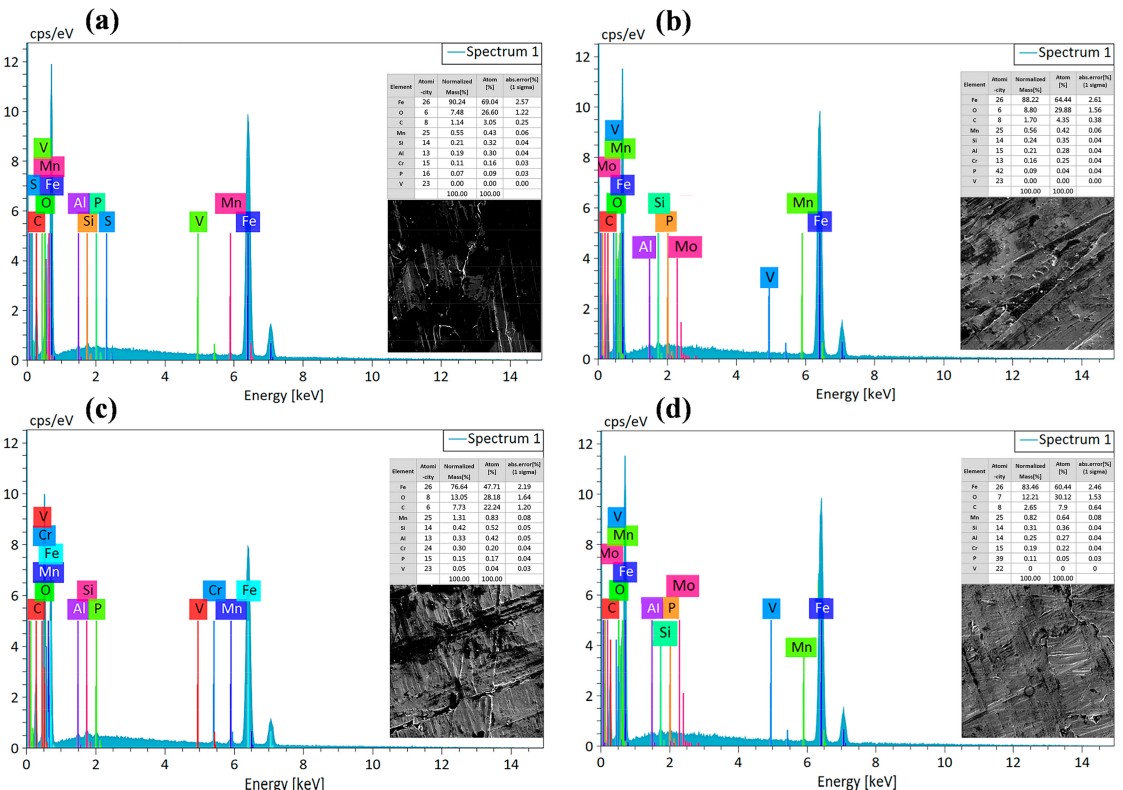

**Figure 5.** Energy spectrum of each sample after different heat treatments: (**a**) H1, (**b**) H2, (**c**) H3, (**d**) H4.

From the above analysis, it can be concluded that the elemental composition of the sample changed slightly after the different heat treatments and that there was a phenomenon of element segregation. Then, the phase of the samples after the different heat treatments and the composition of the particulate matter in Figure 4 were further explored by XRD analysis. Figure 6 displays the XRD diffraction pattern of H13 steel following the various heat treatment processes. The sharp peak shapes observed in Figure 6 for H13 steel after the different heat treatments indicated its favorable crystallization properties following the various heat treatment processes. Moreover, there was little change in the phase composition

under the different heat treatment processes, and there was no austenite diffraction peak, which might be due to the small volume content of the retained austenite after the different heat treatments [14,15]. Through comparison, it could be concluded that the four samples under the different heat treatment processes were mainly composed of a α-Fe phase structure.

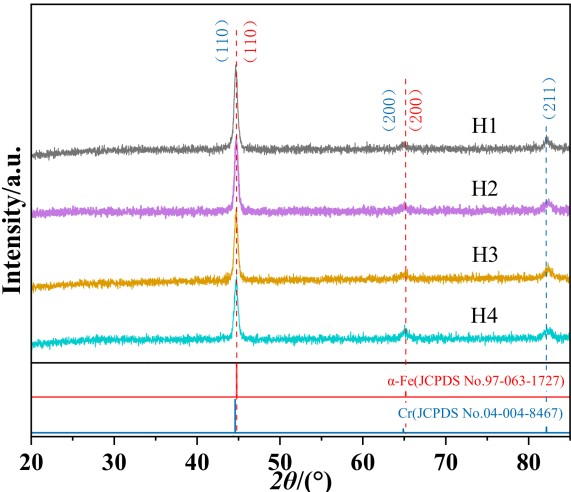

**Figure 6.** Comparison of X-ray diffraction peaks of samples with those of reference materials after different heat treatment processes.

In Figure 6, there are two characteristic peaks of α-Fe (JCPDS No. 97-063-1727): characteristic peak (110), whose diffraction angle is $2\theta_1 = 44.765°$, and characteristic peak (200), whose diffraction angle is $2\theta_2 = 65.166°$. It can be seen from Table 2 that for the samples under the four heat treatment processes, the X-ray diffraction peaks exhibited a leftward shift in comparison to the standard α-Fe diffraction peaks. This was due to the fact that the atomic radii of Cr, V, Mn, Mo, and the other alloy elements were larger than that of the Fe atoms. When the atoms of the above elements form a solid solution with α-Fe, the atoms with larger radii caused lattice distortion in the original lattice. By the Bragg formula:

$$2d\sin\theta = n\lambda \tag{1}$$

**Table 2.** X-ray diffraction peak positions of samples after different heat treatment processes.

| Technology | $2\theta_1$ (110) | $2\theta_2$ (200) | $2\theta_3$ (211) |
| --- | --- | --- | --- |
| H1 | 44.622 | 64.869 | 82.135 |
| H2 | 44.738 | 65.024 | 82.406 |
| H3 | 44.661 | 65.140 | 82.290 |
| H4 | 44.757 | 64.985 | 82.174 |

In the formula: d is the interplanar spacing; θ is the half-diffraction angle; n is the integer; λ is the wavelength of incident X-rays.

It could be concluded that when the lattice constant increased due to the original lattice distortion, there was an increase in the interplanar spacing d. Consequently, the half-diffraction angle θ decreased, leading to a leftward shift in the peak position. In addition, due to the rapid transformation of austenite to martensite in the sample during heat treatment, some carbon remained in the α-Fe matrix during the transformation process, that is, a solid solution in which carbon was supersaturated in α-Fe was formed; this would cause further distortion of the lattice constant of α-Fe.

A characteristic peak with weak signal intensity appeared at 2θ = 82.139°. Through comparative analysis of the XRD standard cards, it could be concluded that the phase might have been dispersed chromium (JCPDS No. 04-004-8467). The characteristic peaks of its phase of matter are (110), (200), and (211), which correspond to the angles $2\theta_1 = 44.765°$,

2θ2 = 44.765°, and 2θ3 = 82.139°, respectively. From the XRD diffraction pattern and Table 2, it is evident that the diffraction peaks of the samples after the various heat treatments correspond well with the peaks of chromium. However, a noticeable phenomenon is observed where these peaks exhibited a rightward shift compared to the standard peaks of chromium. Because chromium and carbon easily form carbides, the carbon atoms with a smaller radius led to the contraction of the chromium lattice, which made its lattice constant smaller. The analysis of formula (1) reveals that there was a decrease in the crystal plane spacing d and an increase in the diffraction angle. These changes resulted in a rightward shift in the position of the chromium diffraction peak. There are no other obvious characteristic peaks appearing in Figure 6. This was due to the small content of other alloying elements in the steel or their low diffusivity in the matrix. Most of the alloying elements formed a substitution solid solution with the $\alpha$-Fe in the form of solutes [16,17].

Combined with the SEM micrographs in Figures 4 and 5, it can be concluded from the above analysis that although there was segregation of the elements C and Cr in the samples after the different heat treatments, no compound of C-Cr was formed, and the particulate matter formed by mixing C and Cr in Figure 4 was a mechanical mixture of C-Cr.

Figure 7 shows the metallographic photo of H13 steel after the different heat treatment processes. A small amount of fine particulate matter was distributed on sample H1 matrix without cryogenic treatment, as shown in Figure 7a. Figure 7b illustrates the presence of a significant amount of finely dispersed granular substances on the surface of the substrate in sample 2 following vacuum gas quenching and cryogenic treatment. This occurrence can be attributed to the promotion of retained austenite decomposition and its subsequent transformation into martensite, which was facilitated by the cryogenic treatment [18]. In addition, due to the decrease in temperature, the solubility of the alloying elements in the matrix decreased, and the alloying elements precipitated to form more fine particles [15,18–20].

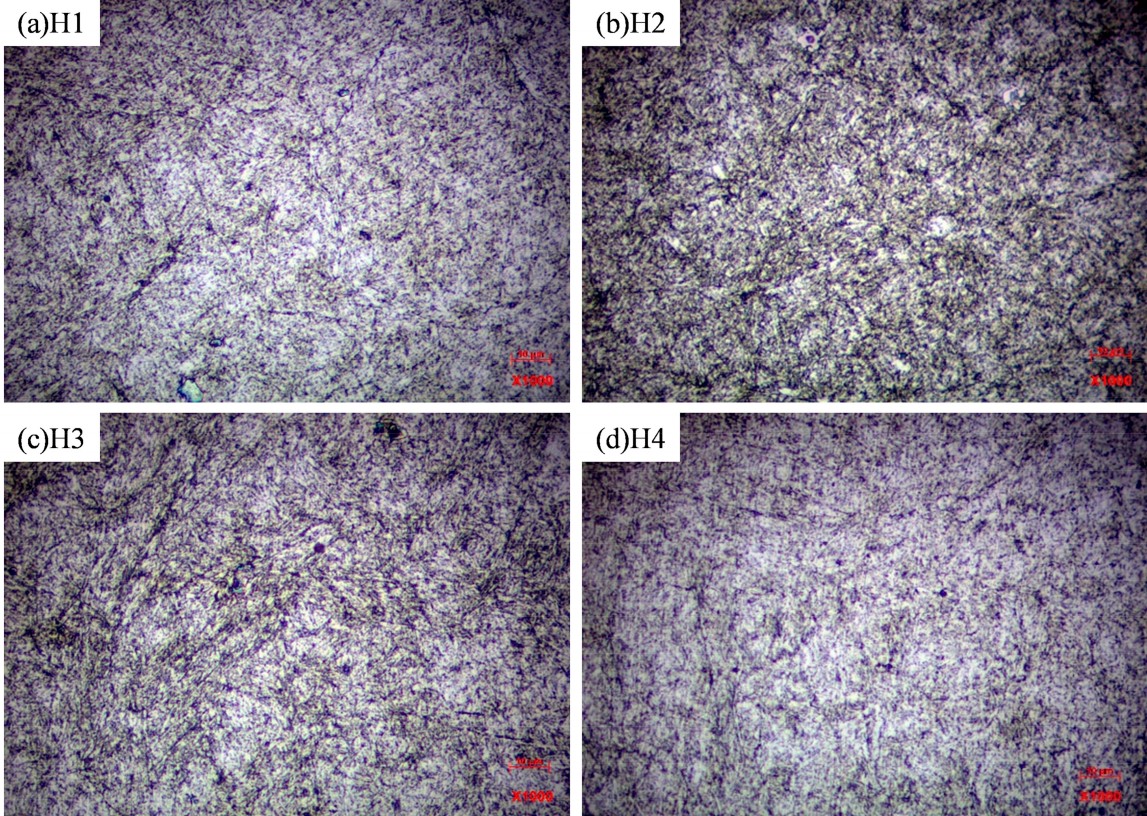

**Figure 7.** Microstructure of H13 steel after different heat treatment processes (×1000): (**a**) vacuum air quenching–tempering; (**b**) vacuum air quenching–cryogenic–tempering; (**c**) vacuum oil quenching–cryogenic–tempering; (**d**) spheroidizing annealing–vacuum air quenching–cryogenic–tempering.

Sample H3 experienced a rapid cooling rate, causing its austenite structure to rapidly cool below the $M_s$ temperature point. This rapid cooling promoted the process of martensite transformation, resulting in the formation of lath martensite with a very small amount of retained austenite within the structure [21]. Sample H2 was cooled with supercooled nitrogen, and the cooling effect became worse with the increase in cooling time; the transformation of supercooled austenite to martensite appeared to exhibit hysteresis, and part of the austenite transformed into a sorbite structure; at the same time, the slower cooling speed led to increased austenite stability [21,22]. According to the XRD analysis, there was no austenite diffraction peak in the different samples, which indicated that the content of retained austenite in all the samples was less. Because the rapid cooling rate would promote the transformation of retained austenite to martensite [23], and the alloying elements would precipitate or segregate at the defects during the cooling process, more alloy elements would precipitate in the subsequent tempering process, forming more strengthening sites and refining the structure [20].

Prior to quenching, sample H4 underwent a spheroidizing process. This spheroidizing annealing facilitated the transformation of flake carbide into spheroidal carbide, which demonstrated lower solubility in austenite compared to flake carbide. Consequently, the presence of spheroidal carbide hindered grain growth within the matrix [24,25]. Tempering causes alloying elements to precipitate and form granular substances, resulting in the formation of a dispersion distribution of hard particle points. It can be seen from Figure 7d that there were more fine hard particles in the microstructure of the spheroidizing annealed sample, which were uniformly dispersed in the tempered martensite matrix, and that the microstructure was denser.

Figure 8 shows the SEM images of the samples under different heat treatment processes. It can be seen from Figure 8 that the fine hard particles were distributed on the matrix of the different samples after the four different heat treatment processes. There were relatively few hard particles in the matrix of sample H1, and a large number of alloying elements still remained in the tempered martensite matrix. The deep cooling of sample H2 facilitated the precipitation of alloying elements, leading to the formation of an increased number of hard particles. These hard particles exhibited a diffuse distribution throughout the tempered martensite matrix, imparting lath-like characteristics [26]. After cryogenic treatment, the content of retained austenite decreased and the content of martensite increased. The cryogenic treatment caused the quenched martensite to undergo lattice deformation and increased dislocations, and the twin grain boundaries increased, resulting in an increase in the supersaturation of carbon atoms, which enhanced the driving force for the precipitation of carbon atoms; thus, a large amount of carbon was precipitated from the martensitic matrix [27]. According to the related research, under the deep cooling condition the whole structure of the material bore a high shrinkage pressure, which promoted the formation of new sub-grain boundaries in the retained austenite grains. At the same time, the newly formed twins and newly formed dislocation lines in the matrix forced the carbon and alloy elements to gather towards the grain boundaries or dislocations. However, at a low temperature, the activity of the carbon atoms was small, and the aggregation phenomenon was more obvious, which provided a recrystallization site for the subsequent tempering process. In the subsequent tempering process, the residual compressive stress played a role in promoting the precipitation of fine carbon alloy particles [28,29]. At a lower temperature, the diffusion ability of the carbon atoms in the material was reduced, leading to a shorter diffusion distance. As a result, the carbon atoms were sporadically dispersed on the surface of the martensite matrix. Furthermore, decarburization caused a transformation of the martensite morphology from needle-like structures to lath-like structures. After subsequent tempering at a higher temperature, the carbon content of the martensite was reduced again, and chromium element was precipitated again, forming finer particles with the precipitated carbon, which were evenly distributed between the tempered martensite laths [30]. Based on the cryogenic treatment, the lattice constant of the martensite was changed; the fragmentation phenomenon of the martensite occurred, and its size was

reduced, as shown in Figure 8b [3]. At the same time, the microstructure of the samples treated by the cryogenic treatment was denser and finer because martensite decomposes, and the lath structure was finer in the subsequent tempering process of the samples treated by cryogenic treatment. At the same time, the surface smoothness of the substrate was also improved by cryogenic treatment.

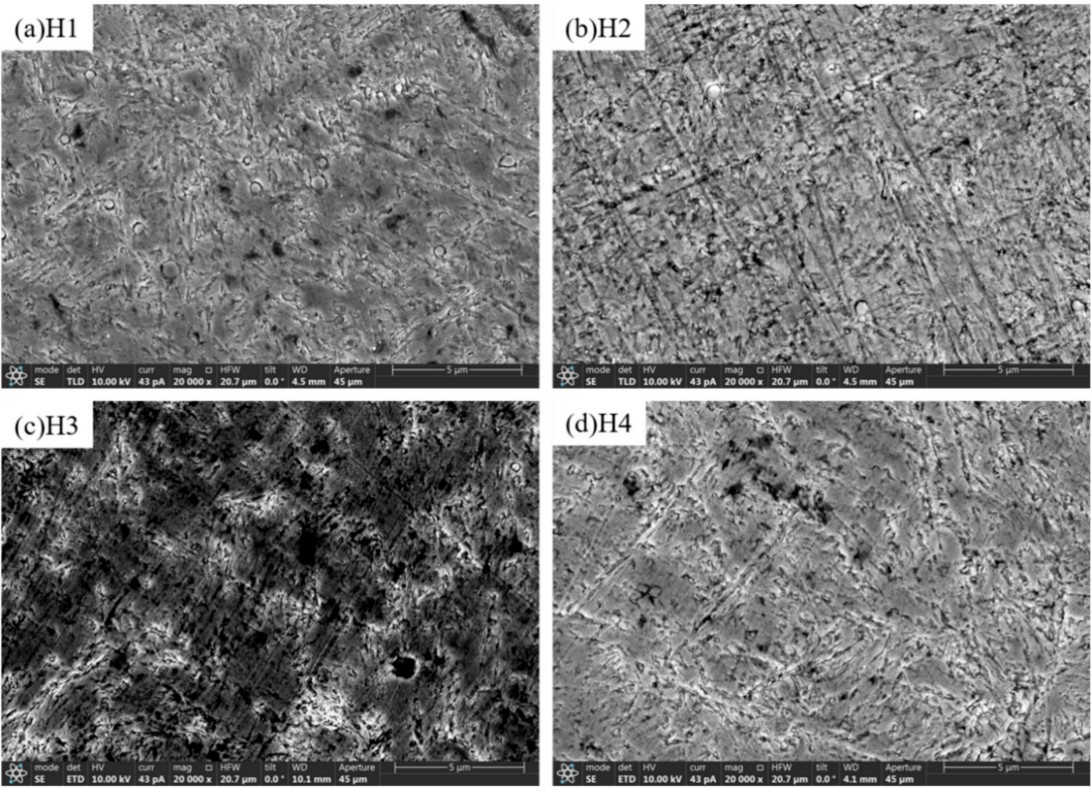

**Figure 8.** SEM photos of H13 steel treated by different heat treatment processes: (**a**) vacuum air quenching–tempering (**b**) vacuum air quenching–cryogenic–tempering (**c**) vacuum oil quenching–cryogenic (**d**) spheroidizing annealing–vacuum air quenching–cryogenic–tempering.

Compared with sample H2, the tempered martensite in sample H3 after vacuum oil quenching was more uniform and finer than that after gas quenching. This phenomenon could be attributed to the faster quenching and cooling rate of sample H3, leading to a significant degree of undercooling. The substantial undercooling served as the driving force for the transformation of austenite into martensite. Furthermore, the increased precipitation of carbon atoms occurred within the quenched martensite matrix [19]. And the supercooled austenite was quickly cooled below the Ms point temperature, which increased the nucleation rate of the martensite. Therefore, the lath-like structure was smaller, as can be seen in Figure 8c. In the subsequent tempering process, the solid-soluble chromium atoms were precipitated and formed mechanical mixed particulate matter with a large number of precipitated carbon atoms. At the same time, the carbide decomposed during the tempering process, and more carbon atoms moved to the dislocation and precipitate. A large number of carbon atoms at the dislocation acted as nucleation sites for recovery and recrystallization during tempering, which further promoted grain refinement during tempering [27].

As can be seen from Figure 8d, the flaky structure of sample H4 was reduced, the spherical structure was increased, and the dispersion degree of the fine C-Cr mechanical mixed particles was further improved. It could be concluded that spherical pearlite and dispersed particulate matter were formed in the matrix after spheroidizing annealing [31]. Moreover, the spherical substances were more difficult to fuse in austenite than flaky substances. This property not only hindered grain growth, it also provided numerous

crystallization cores during the quenching process. As a result, the quenched structure was refined along with the grains [32]. After the tempering process, the C-Cr mechanical mixed particles were fully decomposed, and more carbon atoms were precipitated from the quenched martensite matrix at the same time, which reduced the supersaturation of carbon in the matrix, eliminated a large number of dislocations originally existing in the lath martensite, and made the structure more uniform.

### 3.2. Effect of Heat Treatment Process on Impact Toughness and Hardness

The impact toughness and hardness values of the samples after the different heat treatments are shown in Table 3. The data presented in Table 3 demonstrate that the hardness of the material ranged between 57 and 59 HRC, showing only a small variation as a result of the change in the heat treatment process. The impact toughness of the materials varied greatly and was between 40 J and 100 J. It could be concluded that the heat treatment process had a significant effect on impact toughness. Figure 9 shows the hardness, impact work, and corresponding standard deviation of the samples under the different heat treatment processes. Among the samples, sample H1 exhibited the highest hardness but the lowest impact energy, whereas sample H4 demonstrated the highest impact energy along with a higher hardness. It can be concluded from Figure 9 that the hardness and impact work of the materials showed opposite trend under the different heat treatment processes.

**Table 3.** Hardness and impact work of improved H13 under different heat treatment processes.

| Heat-Treated Sample | Relative Position of Sample | Hardness Value/HRC | | | | Average Hardness/HRC | Hardness Standard Deviation | Impact Energy/J | | | | Average Impact Energy/J | Standard Deviation of Impact Energy |
|---|---|---|---|---|---|---|---|---|---|---|---|---|---|
| | | 1 | 2 | 3 | 4 | | | 1 | 2 | 3 | 4 | | |
| H1 | Upper left | 58.5 | 59.5 | 58 | 58 | 58.5 | 0.707 | 39 | 38.5 | 41.5 | 41.5 | 40.1 | 1.601 |
| | Upper right | 59 | 58 | 57.5 | 60 | 58.63 | 1.109 | 52.5 | 56 | 43 | 42 | 48.4 | 6.945 |
| | Bottom left | 58 | 58 | 59.5 | 58.5 | 58.5 | 0.707 | 44 | 40 | 44 | 40 | 42.0 | 2.309 |
| | Bottom right | 58 | 58 | 59 | 58 | 58.25 | 0.5 | 72 | 60 | 47 | 47 | 56.5 | 12.014 |
| | Average | 58.38 | 58.38 | 58.5 | 58.63 | 58.47 | | 51.88 | 48.63 | 43.88 | 42.63 | 46.75 | |
| H2 | Upper left | 57.5 | 57.5 | 57 | 58 | 57.50 | 0.408 | 62 | 77 | 70 | 74 | 70.75 | 6.5 |
| | Upper right | 57.5 | 57 | 58 | 57.5 | 57.50 | 0.408 | 85 | 67 | 67 | 56 | 68.75 | 12.010 |
| | Bottom left | 57.5 | 57.5 | 58 | 58 | 57.75 | 0.289 | 68 | 78 | 78 | 70 | 73.50 | 5.260 |
| | Bottom right | 57 | 57.5 | 57 | 57 | 57.13 | 0.25 | 90 | 63 | 61 | 57 | 67.75 | 15.042 |
| | Average | 57.38 | 57.38 | 57.5 | 57.63 | 57.47 | | 76.25 | 71.25 | 69 | 64.25 | 70.19 | |
| H3 | Upper left | 58.5 | 58 | 59 | 59 | 58.38 | 0.479 | 80 | 75 | 70 | 78 | 75.75 | 4.349 |
| | Upper right | 58.5 | 58.5 | 58 | 59 | 58.50 | 0.408 | 64 | 56 | 59 | 80 | 64.75 | 10.689 |
| | Bottom left | 58.5 | 58 | 58 | 59 | 58.25 | 0.479 | 88 | 68 | 88 | 60 | 76.00 | 14.236 |
| | Bottom right | 58.5 | 58.5 | 58 | 59 | 58.38 | 0.408 | 64 | 82 | 86 | 90 | 80.50 | 11.475 |
| | Average | 58.5 | 58.25 | 58.25 | 59 | 58.38 | | 74 | 70.25 | 75.75 | 77 | 74.25 | 58.5 |
| H4 | Upper left | 58 | 58 | 58.5 | 58 | 58.13 | 0.25 | 57 | 88 | 106 | 138 | 97.25 | 33.876 |
| | Upper right | 57.5 | 58 | 58 | 58 | 57.88 | 0.25 | 92 | 80 | 84 | 98 | 88.50 | 8.062 |
| | Bottom left | 57.5 | 57.5 | 58 | 58 | 57.75 | 0.289 | 66 | 110 | 72 | 61 | 77.25 | 22.292 |
| | Bottom right | 58 | 57.5 | 57.5 | 57.5 | 57.63 | 0.25 | 108 | 82 | 74 | 75 | 84.75 | 15.903 |
| | Average | 57.75 | 57.75 | 58 | 57.88 | **57.85** | | 80.75 | 90 | 84 | 93 | 86.94 | 57.75 |

From Figure 9a and Table 3, the analysis reveals that sample H4 exhibited the smallest standard deviation in hardness value, suggesting that its hardness value displayed minimal fluctuation and demonstrated superior stability. The standard deviation of the hardness value of sample H1 was the largest, which indicated that the hardness value of this material had the highest degree of dispersion and poor hardness uniformity. The standard deviation of the impact energy of the material obtained by sample H1 was the smallest, which indicated that the toughness of the material was more uniform, but its toughness was low. The obtained data show that sample H4 had the largest standard deviation in the impact work, indicating that its toughness value exhibited the highest degree of dispersion. Furthermore, this suggests an uneven distribution of toughness within the material.

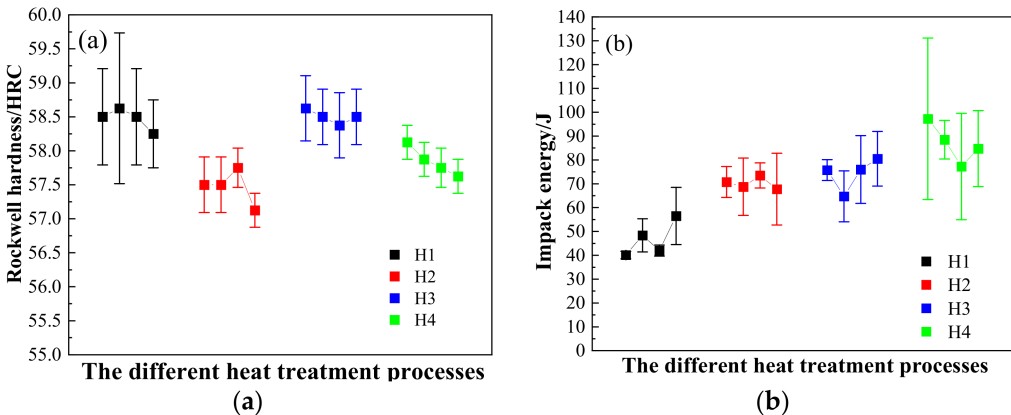

**Figure 9.** Hardness (**a**), impact work (**b**), and corresponding standard deviation of samples under different heat treatment processes.

As observed in Figure 9 and Table 3, it is evident that sample H2 exhibited a slight decrease in hardness from 58.47 HRC to 57.47 HRC when compared to sample H1. However, in contrast, the impact toughness of sample H2 increased significantly from 46.75 J to 70.19 J, showing an increase of approximately 50%. And the C-Cr mechanical mixed particles distributed on the matrix of sample H2 were much smaller and more evenly distributed. This might be because cryogenic treatment promotes the transformation of retained austenite into martensite, and more carbon atoms were precipitated from the quenched martensite matrix. After the carbon atoms were precipitated, the martensite changed from a hard and brittle needle-like substance to a high-toughness plate-like substance due to the decrease in carbon content, as shown in Figure 8b. Deep cooling caused the carbon atoms in the quenched martensite to segregate. In the presence of distortion stress, the supersaturated carbon atoms tended to segregate towards defects, resulting in the formation of finer granular substances together with precipitated chromium atoms (refer to Figures 7b and 8b). Simultaneously, there was a possibility of segregated carbon atoms diffusing into the retained austenite that had not undergone martensite transformation, thereby enhancing the stability of the retained austenite. And after cryogenic treatment, the residual austenite was in an equiaxed compressive stress state, and the plastic deformation tendency disappeared; it appeared as a tough phase in use, which could alleviate stress, and it prevented the contact fatigue from expanding. As a result, this process improved the impact toughness of steel [33]. Moreover, after cryogenic treatment, the content of martensite increased, and the number of martensite interfaces increased. During the tempering process, martensite underwent recovery and recrystallization, leading to the formation of a finer structure and a more uniform distribution of C-Cr mechanical mixed particles. This effect was advantageous in reducing the brittleness and residual stress of hardened steel. When impacted, the material could absorb more impact energy, which further improved the toughness of the material [34]. Studies have shown that the hardness of steel is improved only when it is tempered at low temperature after cryogenic treatment. When the temperature exceeded that of low-temperature tempering, the hardness of the sample was lower in comparison to the sample without cryogenic treatment. This finding aligned with the conclusion drawn in this paper regarding the lower hardness of sample H2 compared to sample H1 [35].

The hardness of sample H2 quenched by vacuum gas was lower than that of sample H3 quenched by vacuum oil. This was because the cooling rate of sample H2 was slower than that of oil quenching. A slower cooling rate decreased the driving force for the transformation of austenite into martensite. This slower cooling rate also led to the occurrence of the hysteresis phenomenon during the martensite transformation process. Consequently, the stability of the austenite phase was enhanced, resulting in a decrease in the hardness of the microstructure. Simultaneously, the vacuum oil quenching process effectively achieved

a faster cooling rate. This accelerated cooling rate facilitated the precipitation of super-saturated carbon atoms at a rapid pace. Consequently, a distorted dipole stress field was formed around the carbon atoms at the center, leading to an increased dislocation density within the structure. Moreover, the dislocations exhibit a strong interaction with the stress field [36]. Therefore, the microhardness of sample H3 reached 58.38 HRC. Combined with the XRD analysis of the different samples under the two processes, it could be concluded that the difference in the retained austenite content was not big, and the difference in sample hardness was mostly caused by the different number and state of the hard particles. According to related research, the fracture toughness of steel materials mostly depends on the content of retained austenite. When the difference in retained austenite content is not big, the more fine hard particles there are, the more uniform the distribution is, and the better the toughness of the material is [20]. During the subsequent tempering treatment, the final microstructure of the two samples was different due to the difference in the parent phases. Based on the analysis of the metallographic diagram and SEM photos, it can be observed that the C-Cr mechanical mixed particles, which were distributed in a dispersed manner following the oil quenching process, exhibited a finer size and were distributed uniformly on the ferrite matrix. These particles served to reinforce the material as a second phase strengthening mechanism. The dispersed distribution of the second phase hindered the movement of the grain boundaries and dislocations and improved the strength of the material. At the same time, from a macro point of view, vacuum oil quenching would promote the surface carburization of the materials and further improve the hardness of the materials. The microstructure image reveals that the tempered martensite lathing present in the microstructure of sample H3 was more pronounced and exhibited a finer structure. As a result, there was a notable enhancement in its impact performance, with a notable increase to 74.25 J. Combining various strengthening mechanisms with the more prominent lathing of the matrix, sample H3 after oil quenching had both high hardness and high toughness [37].

Following the spheroidizing annealing process, sample H4 demonstrated preferable hardness uniformity, as indicated by its having the lowest hardness standard deviation. Furthermore, there was a slight increase in the hardness value to 57.85 HRC. Most notably, the impact energy significantly improved to 86.94 J, representing a 23.8% increase compared to the non-spheroidized sample H2. Metallographic and SEM observation show that the tempered martensite matrix of sample H4 was finer after spheroidizing annealing, and the finer the microstructure, the greater the force between the dislocations retained after tempering, which led to the increase in the overall hardness of the sample [38]. However, the dislocation had orientation, which was also an important factor leading to the large standard deviation of the impact work of its samples. The heating and heat preservation process during spheroidizing annealing facilitated the formation of a greater quantity of finely dispersed spherical C-Cr mechanical mixed particles. Subsequently, during the preheating and quenching process, a quenched martensite phase with a refined structure was generated [39]. After high-temperature tempering three times, more alloy elements were dissolved in the tempered martensite matrix, which improved the tempering stability. Fine C-Cr mechanical mixed particles dispersed in the matrix led to secondary hardening, which further strengthened the hardening effect [40].

### 3.3. Effect of Heat Treatment Process on Wear Resistance

Figure 10a and Table 4 show the cumulative wear of H13 steel after the different heat treatment processes. The analysis of Figure 10a and Table 4 reveals that sample H1 exhibited the lowest cumulative wear, while sample H2 had the highest cumulative wear. Furthermore, the observed trend in wear was consistent with the corresponding trend in hardness. As can be seen from Figure 10a and Table 4, the wear of H13 steel increased following cryogenic treatment. This could be attributed to the promotion of the transformation of retained austenite to martensite through cryogenic treatment. At extremely low temperatures, the martensite underwent lattice deformation, leading to

increasing supersaturation of the carbon atoms. Consequently, carbon segregation occurred at the defects, ultimately enhancing the stability and comprehensive mechanical properties of the structure. At the same time, because a large number of carbon atoms were dissolved at an extremely low temperature, they formed fine hard particles with alloy elements. The carbon content of the martensite after cryogenic treatment decreased. It led to the increase of lath martensite, the decrease of microstructure hardness and wear resistance. During the tempering process of lath martensite, the carbon atoms had a tendency to aggregate at the dislocations and crystal planes. This aggregation subsequently resulted in the precipitation of numerous fine hard particles along the grain boundaries within the tempered structure. As a consequence, this precipitation weakened the cohesion among the matrix grains to some extent, thereby compromising the strength of the grain boundaries and reducing the wear resistance of the material. The wear resistance of sample H3 was better than that of sample H2, which was the result of a microstructural change due to the different cooling rates. The cooling rate was high, the time for the diffusion of elements was short, and the diffusive phase transition had difficulty occurring. With the increase in cooling rate, the diffusion speed of the elements was slower, which made the energy required for carbon nucleation insufficient or too late to allow nucleation. Ultimately, a portion of carbon and a significant amount of the alloy elements that were initially dissolved in austenite existed in the tempered martensite matrix in the form of a solid solution This solid solution imparts solid solution strengthening, resulting in improved material hardness and enhanced wear resistance [41]. Concurrently, the analysis of the microstructure indicated that the tempered martensite matrix structure of sample H3, following oil quenching, exhibited a finer grain structure. Additionally, a greater quantity of finely dispersed hard particles was observed on the matrix, thereby contributing to second phase strengthening. Consequently, these findings collectively resulted in an overall enhancement of the material's wear resistance [42]. There would be hysteresis in the cooling process of undercooled austenite, resulting in more retained austenite in the quenched structure. Due to the lack of a cooling rate, some austenite would be transformed into sorbite. During the subsequent tempering process of this sample, due to the presence of sorbite, the unevenness of the final structure increased, which was another reason for the large deviation in impact energy. The structure of sample H4, after spheroidizing annealing, exhibited a finer grain structure. In contrast, the sample without spheroidizing annealing displayed larger hard particles of C-Cr mechanical mixed particles on the surface of the matrix. Based on the existence of larger particles, the contact area between them and the matrix was larger, and the interface might be the site where potential cracks originate, which might be the main reason for the decrease in wear resistance of sample H2. During the spheroidizing annealing process of sample H4, when the austenite slowly cooled from a high temperature to a certain extent, the carbon content reached saturation before the eutectoid transformation. Cementite and carbon atoms would precipitate along the austenite grain boundaries during the continuous cooling process. Spherical C-Cr mechanical mixed particles exhibited low solubility in austenite, leading to the formation of numerous hard particles in the quenched microstructure. After subsequent tempering, the C-Cr mechanical mixed particles retained in the quenched martensite were fully melted, and the spherical hard particles originally remaining on the quenched martensite matrix served as nucleation sites while melting, forming more spherical hard particles, which were uniformly distributed on the surface of the matrix. And the second phase strengthening effect was obvious. The circular geometry of the smaller C-Cr mechanical mixed particles resulted in a reduced contact area with the matrix, thus enhancing the crack resistance (improved impact energy) of sample H4. Additionally, the microstructure of sample H4 exhibited greater uniformity, and the enhanced densification contributed to its improved wear resistance. As can be seen from Figure 10b and Table 5, the unit wear of sample H1 and sample H4 all showed the same continuous downward trend. However, its strengthening mechanism was different. Sample H1 had the highest hardness, resulting in good wear resistance. Sample H4 had a good match between strength and toughness based on

secondary strengthening, and its wear change was the most gentle, which showed that its structure was more compact and uniform. The wear of samples H2 and H3 experienced a significant increase in the later stages of wear, which was potentially attributable to the presence of a thin hard layer within these samples. Under continuous wear conditions, the rapid exposure of the soft internal structure led to a reduction in their wear resistance.

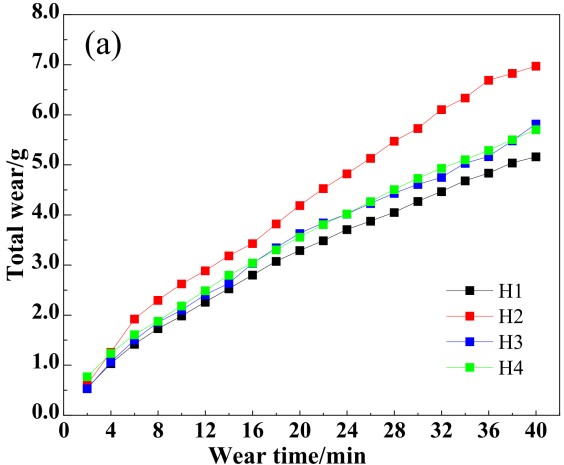 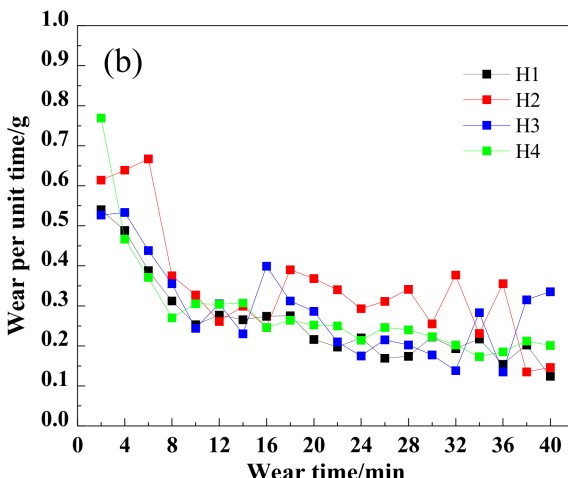

**Figure 10.** (**a**) total wear amount and (**b**) wear amount per unit time of H13 steel after different heat treatments.

**Table 4.** Total wear of improved H13 under different heat treatment processes.

| Wear Time | H1 | | H2 | | H3 | | H4 | |
|---|---|---|---|---|---|---|---|---|
| | Average Value | Standard Deviation | Average Value | Standard Deviation | Average Value | Standard Deviation | Average Value | Standard Deviation |
| 2 (min) | 0.54 | 0.01732 | 0.614 | 0.00265 | 0.527 | 0.00173 | 0.769 | 0.00361 |
| 4 | 1.028 | 0.00173 | 1.253 | 0.00173 | 1.06 | 0.002 | 1.236 | 0.00173 |
| 6 | 1.416 | 0.001 | 1.92 | 0.002 | 1.498 | 0.00265 | 1.607 | 0.00173 |
| 8 | 1.728 | 0.00173 | 2.295 | 0.00173 | 1.853 | 0.00173 | 1.877 | 0.002 |
| 10 | 1.981 | 0.001 | 2.622 | 0.00173 | 2.097 | 0.00265 | 2.182 | 0.00173 |
| 12 | 2.258 | 0.00361 | 2.883 | 0.001 | 2.402 | 0.00173 | 2.486 | 0.00265 |
| 14 | 2.523 | 0.002 | 3.182 | 0.002 | 2.632 | 0.00265 | 2.793 | 0.00173 |
| 16 | 2.797 | 0.002 | 3.428 | 0.002 | 3.031 | 0.00173 | 3.039 | 0.00265 |
| 18 | 3.072 | 0.00265 | 3.818 | 0.002 | 3.343 | 0.00173 | 3.303 | 0.00265 |
| 20 | 3.288 | 0.00173 | 4.186 | 0.00361 | 3.629 | 0.001 | 3.555 | 0.00265 |
| 22 | 3.485 | 0.00173 | 4.526 | 0.002 | 3.839 | 0.00265 | 3.805 | 0.42464 |
| 24 | 3.705 | 0.00173 | 4.819 | 0.00265 | 4.014 | 0.00436 | 4.019 | 0.00173 |
| 26 | 3.874 | 0.00265 | 5.13 | 0.00265 | 4.229 | 0.00265 | 4.265 | 0.001 |
| 28 | 4.048 | 0.003 | 5.471 | 0.001 | 4.431 | 0.00173 | 4.505 | 0.003 |
| 30 | 4.27 | 0.00265 | 5.726 | 0.002 | 4.608 | 0.00265 | 4.728 | 0.00173 |
| 32 | 4.463 | 0.002 | 6.103 | 0.00173 | 4.746 | 0.00173 | 4.93 | 0.02 |
| 34 | 4.68 | 0.002 | 6.334 | 0.00361 | 5.029 | 0.003 | 5.103 | 0.00265 |
| 36 | 4.834 | 0.00265 | 6.689 | 0.003 | 5.164 | 0.00173 | 5.288 | 0.00265 |
| 38 | 5.036 | 0.00265 | 6.824 | 0.00173 | 5.479 | 0.00265 | 5.5 | 0.01 |
| 40 | 5.16 | 0.00265 | 6.97 | 0.00173 | 5.814 | 0.00173 | 5.701 | 0.00265 |

**Table 5.** Unit wear of improved H13 under different heat treatment processes.

| Wear Time | H1 | | H2 | | H3 | | H4 | |
|---|---|---|---|---|---|---|---|---|
| | Average Value | Standard Deviation | Average Value | Standard Deviation | Average Value | Standard Deviation | Average Value | Standard Deviation |
| 2 (min) | 0.54 | 0.02646 | 0.614 | 0.002 | 0.527 | 0.00265 | 0.769 | 0.00265 |
| 4 | 0.488 | 0.00265 | 0.639 | 0.00346 | 0.533 | 0.00819 | 0.467 | 0.002 |
| 6 | 0.388 | 0.001 | 0.667 | 0.00173 | 0.438 | 0.00265 | 0.371 | 0.00173 |
| 8 | 0.312 | 0.00173 | 0.375 | 0.00265 | 0.355 | 0.00265 | 0.27 | 0.002 |
| 10 | 0.253 | 0.00361 | 0.327 | 0.00265 | 0.244 | 0.001 | 0.305 | 0.00265 |
| 12 | 0.277 | 0.00173 | 0.261 | 0.00265 | 0.305 | 0.00346 | 0.304 | 0.00265 |
| 14 | 0.265 | 0.002 | 0.299 | 0.00346 | 0.23 | 0.001 | 0.307 | 0.002 |
| 16 | 0.274 | 0.00173 | 0.246 | 0.002 | 0.399 | 0.00265 | 0.246 | 0.00173 |
| 18 | 0.275 | 0.00173 | 0.39 | 0.00173 | 0.312 | 0.00265 | 0.264 | 0.00173 |
| 20 | 0.216 | 0.00173 | 0.368 | 0.003 | 0.286 | 0.00173 | 0.252 | 0.00173 |
| 22 | 0.197 | 0.001 | 0.34 | 0.00862 | 0.21 | 0.00985 | 0.25 | 0.00265 |
| 24 | 0.22 | 0.001 | 0.293 | 0.00265 | 0.175 | 0.00173 | 0.214 | 0.00265 |
| 26 | 0.169 | 0.001 | 0.311 | 0.00265 | 0.215 | 0.00265 | 0.246 | 0.00173 |
| 28 | 0.174 | 0.00173 | 0.341 | 0.00173 | 0.202 | 0.00173 | 0.24 | 0.00173 |
| 30 | 0.222 | 0.00173 | 0.255 | 0.003 | 0.177 | 0.00173 | 0.223 | 0.00173 |
| 32 | 0.193 | 0.003 | 0.377 | 0.002 | 0.138 | 0.00265 | 0.202 | 0.002 |
| 34 | 0.217 | 0.00173 | 0.231 | 0.00346 | 0.283 | 0.001 | 0.173 | 0.00361 |
| 36 | 0.154 | 0.002 | 0.355 | 0.00265 | 0.135 | 0.00265 | 0.185 | 0.00265 |
| 38 | 0.202 | 0.00361 | 0.135 | 0.00361 | 0.315 | 0.002 | 0.212 | 0.00173 |
| 40 | 0.124 | 0.003 | 0.146 | 0.00173 | 0.335 | 0.00173 | 0.201 | 0.003 |

## 4. Conclusions

(1) The elemental composition of high-carbon H13 steel after different heat treatments did not change much, but there was still segregation of the alloy elements and dispersion of the C-Cr mechanical mixed particles.

(2) After heat treatment, the H13 steel had no diffraction peaks of multiple alloying elements, and its phases were mainly the $\alpha$-Fe phase and Cr phase. The characteristic peaks observed in the XRD patterns of each sample exhibited a small deviation in peak position, which can be attributed to the formation of a solid solution through the incorporation of alloy elements into the matrix.

(3) After the different heat treatment processes, fine C-Cr mechanical mixed particles could be seen dispersed in the ferrite matrix. The analysis revealed that the cryogenic process facilitated the transition from austenite to martensite, while promoting the precipitation of carbon atoms and alloy elements, resulting in the formation of numerous fine hard particles dispersed on the surface of the matrix. At the same time, the retained austenite was divided by lath martensite, and the structure was refined. The oil quenching process provided a large undercooling, promoted the precipitation of carbon atoms, formed more fine particles, improved the nucleation rate of martensite, and further refined the structure. The spheroidizing annealing process was added before the gas quenching cryogenic process to promote the melting of the original carbide and to obtain more dispersed hard particles, which inhibited the growth of the original grains and refined the structure.

(4) Following heat treatment, the sample exhibited a small range of hardness variation, while displaying significant variability in impact toughness. Sample H1 had the highest hardness and the lowest impact toughness. The average hardness of sample H2 was lower than that of sample H1, but its toughness was improved. After oil quenching and cryogenic treatment, the hardness and toughness of sample H3 were improved compared with those without cryogenic treatment. Following the spheroidizing annealing process, sample H4 exhibited elevated hardness and ultra-high toughness. And its special process allowed the material to obtain the best matching between strength and toughness.

(5) After the different heat treatments, the change trend of the wear resistance and hardness was consistent, that is, the higher the hardness, the better the wear resistance. While maintaining high wear resistance, the toughness of sample H4 was notably enhanced, resulting in an improved performance of the material. The process was an ideal heat treatment process for the high-carbon H13 steel used for shield tools.

To sum up, based on the research on the quenching medium of improved H13 steel heat treatment, the preliminary heat treatment mode, and the mechanism of cryogenic treatment, a better heat treatment combination mode of improved H13 steel was obtained, which greatly improved the strength and toughness matching of this steel. In this era of increasingly developed traffic, there will be more and more occasions for shield machine construction in the future, and the performance requirements of the materials used for shield machine tools will be higher and higher. The results obtained in this paper provide some empirical summaries for the toughening and strengthening of H13 steel and provide some insights into the improvement of the application and expansion of H13 steel in the shield field.

**Author Contributions:** Validation, W.F.; Resources, X.L.; Writing—original draft, K.D.; Writing—review & editing, L.X.; Project administration, Z.L.; Funding acquisition, R.Z. All authors have read and agreed to the published version of the manuscript.

**Funding:** This work was supported in part by "Shandong Province Science and Technology Small and Medium Enterprises Innovation Ability Enhancement Project" (No. 2022TSGC2589), "Liaocheng University Project Fund" (No. K23LB0101) and "Liaocheng University Crosswise Tasks" (No. K23LD60).

**Data Availability Statement:** Data are contained within the article.

**Acknowledgments:** The authors are grateful for the facilities and other support given by "Shandong Province Science and Technology Small and Medium Enterprises Innovation Ability Enhancement Project" (No. 2022TSGC2589), "Liaocheng University Project Fund" (No. K23LB0101) and "Liaocheng University Crosswise Tasks" (No. K23LD60). We would like to thank them for the financial support.

**Conflicts of Interest:** Author Lipeng Xu was employed by the company Shandong EAST Engineering Tools Limited Liability Company. The remaining authors declare that the research was conducted in the absence of any commercial or financial relationships that could be construed as a potential conflict of interest.

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
