# Peer review of "Effect of Heat Treatment Process on Microstructure and Mechanical Properties of High-Carbon H13 Steel"

_processes, doi:10.3390/pr11113239_

Round 1

Reviewer 1 Report

Comments and Suggestions for Authors

The manuscript titled "Effect of Heat Treatment Process on microstructure and Mechanical Properties of High Carbon H13 Steel" describes influence of the heat treatment parameters on the H13 Steel properties. The study involves very complex heat treatment technologies and obtained results could be very valuable for further studies related to tunneling shield construction. The paper should undergo revision of the text before being able for acceptance in Processes Journal. 

The recommendations for text editing:

1. Abstract is overloaded with results, and it should be revised in a way to clearly indicate the most important results. Please remove less important results.

2. In the section 2 "test materials and methods" should be replaced with "Materials and methods".

3. In the section 2, Fig. 1, it would be good to add dots to mark position chosen for hardness test.

4. Table 2.  2θ12θ22θ3, should be replaced with 2θ1(100), 2θ2(200), 2θ3(211).

5. In the Conclusions, it would be good to add one sentence to emphasize importance of the presented results for future research in the field.

Comments on the Quality of English Language

Authors should proofread text, there are quite long sentences which are not easy to follow. Also, there are many repetitive words, sometimes in the same sentence. Please recheck some words or phrases, like "....carbon atoms in the tissue was weaker...". Tissue is not commonly used word in this case, etc. 

Author Response

Dear Reviewer 1

   Thanks for your kind comments. According to your comments, we have revised our manuscript carefully. To let you know our revision, the corresponding replies to your comments have been written as the following:
Question 1. Abstract is overloaded with results, and it should be revised in a way to clearly indicate the most important results. Please remove less important results.

Response: Thanks for your suggestion. The abstract has been revised and some unimportant results have been deleted.

Abstract: This paper investigated the mechanical properties and microstructure of different samples of H13 steel after undergoing various heat treatment processes. It provided a detailed analysis of the microstructure and mechanical properties under different processes, approaching the topic from a theoretical perspective. The phase composition of each sample remained unchanged after undergoing different heat treatment processes. Despite the Vacuum Gas-quenching(H1) sample being guaranteed a hardness of 58.47HRC, its toughness fell below expectations at a mere 46.75J. Notably, the microstructure of the sample which underwent H1 process and cryogenic(H2) treatment exhibited a finer grain size and higher toughness compared to the sample which only underwent H1 process without cryogenic treatment. Its toughness was 70.19J,but its hardness slightly decreased to 57.47HRC. Following the application of Oil-quenching and cryogenic treatment(H3), the hardness of the sample significantly increased, reaching a remarkable 58.38 HRC. Additionally, the sample exhibited good impact resistance, measuring at 74.25 J. Before the H2 process, the sample which underwent spheroidizing annealing process(H4) had a higher hardness compared to the sample without spheroidizing annealing. At the same time, when comparing the above four samples, the sample that underwent H4 process exhibited the best toughness with the value of 86.94 J, while still maintaining a hardness of 57.85 HRC, achieving an ideal balance between strength and toughness. Therefore, the optimal heat treatment process for high-carbon H13 steel was spheroidizing annealing followed by vacuum gas quenching, and then cryogenic treatment.

Question 2. In the section 2 "test materials and methods" should be replaced with "Materials and methods".

Response: Thanks for your careful inspection. The description of " test materials and methods " has been replaced by the "Materials and methods".

Question 3. In the section 2, Fig. 1, it would be good to add dots to mark position chosen for hardness test.

Response: Thanks for your suggestion. For different samples after heat treatment, the same surface was selected as the hardness test surface (size is 55mm×10mm). Taking the geometric center of this surface as the symmetrical center, a rectangular area of 20mm×10mm was selected as the hardness test area, as shown in Figure 2. Each sample randomly selected four points in the rectangular area as hardness test points, and calculated the average and standard deviation of the four measured values.

Figure 2. Hardness test area

Question 4. Table 2. 2θ1, 2θ2, 2θ3, should be replaced with 2θ1(110), 2θ2(200), 2θ 3(211).

Response: Thanks for your careful inspection. The description of " 2θ1, 2θ2, 2θ3 in Table 2 " has been replaced by the "2θ1(110), 2θ2(200), 2θ3(211)".

Table 2. X-ray diffraction peak positions of samples after different heat treatment processes.

Technology

2θ1(110)

2θ2(200)

2θ3(211)

H1

44.622

64.869

82.135

H2

44.738

65.024

82.406

H3

44.661

65.140

82.290

H4

44.757

64.985

82.174

Question 5. In the Conclusions, it would be good to add one sentence to emphasize importance of the presented results for future research in the field.
Response: Thanks for your suggestion. Statements emphasizing the significance of the displayed results for future research in this field have been added to the conclusion paragraph.

To sum up, based on the research on the quenching medium of improved H13 steel heat treatment, the preliminary heat treatment mode and the mechanism of cryogenic treatment, a better heat treatment combination mode of improved H13 steel is obtained, which greatly improves the strength and toughness matching of this steel. In this era of increasingly developed traffic, there will be more and more occasions for shield machine construction in the future, and the performance requirements of materials used for shield machine tools will be higher and higher. The results obtained in this paper provide some empirical summaries for the toughening and strengthening of H13 steel, and provide some insights for improving the application and expansion of H13 steel in shield field

Question 6. Authors should proofread text, there are quite long sentences which are not easy to follow. Also, there are many repetitive words, sometimes in the same sentence. Please recheck some words or phrases, like "....carbon atoms in the tissue was weaker...". Tissue is not commonly used word in this case, etc.

Response: Thanks for your careful inspection. The words, long sentences and grammar in our manuscript have been properly revised.

I sincerely appreciate your kind comments very much.

Best Regards

Sincerely Yours

Lipeng Xu

Nov. 3, 2023

Reviewer 2 Report

Comments and Suggestions for Authors

This paper investigated the mechanical properties and microstructure of different samples of H13 steel after undergoing various heat treatment processes. It provided a detailed analysis of the microstructure and mechanical properties under different processes, approaching the topic from a theoretical perspective. The composition of each sample remained unchanged after undergoing different heat treatment processes.

THE WHOLE WORK IS INTERESTING AND COULD HAVE POTENTIAL APPLICATIONS.

POINTS FOR IMPROVEMENT :

1. Please, provide the manufacturer and the country of the instruments used in the analysis.

2. As far as I can see, there is not a basis for properties for the measured properties. In other words the provided steel by the manufacturer could be further analyzed before any treatment and the effect of any treatment could be also compared to this basis. Alternatively, you could provide measurements from the literature as a basis for comparison.

Author Response

Dear Reviewer 2

   Thanks for your kind comments. According to your comments, we have revised our manuscript carefully. To let you know our revision, the corresponding replies to your comments have been written as the following:
Question 1. Please, provide the manufacturer and the country of the instruments used in the analysis.

Response: Thanks for your suggestion. Information about the manufacturer and country of the instruments used in the experimental analysis has been added to the manuscript.

The room temperature impact specimens were selected according to their positions relative to the axis of the cutter ring, which were named outer axis, outer diameter, inner diameter, and inner axis. The samples were all were all processed by wire cutting + grinding machine, and the unnotched standard specimens were prepared according to ISO 148-1-2016 carbon steel impact test standard. As shown in Fig. 1, its size was 55mm×10mm×7mm, and the impact test was carried out on JB-300B pendulum testing machine produced by China Jinan Time Testing Instrument Co., Ltd. Using Zeiss optical microscope manufactured by Carl Zeiss CMP GmbH in Germany and scanning electron microscope to observe the microstructure after different heat treatments, the etchant used for microstructure observation is 3% nitric acid alcohol and test standard was implemented in accordance with ISO 945-1-2008. After the sample underwent grinding and polishing, the material phase was analyzed by Brooke D8 advanced X-ray powder diffractometer manufactured by Brook AXS Co., Ltd., Germany, and the scanning angle range was 20 to 85 according to the relevant standards of ISO 21068-2023. Using HR-150A hardness tester produced by China Nanjing Wode Analytical Instrument Manufacturing Co., Ltd., the samples were tested for Rockwell hardness. The test was conducted in accordance with ISO 6508-4-2018.The samples after different heat treatments were cut and sampled, the size was 40mm×20mm×20mm, and the same orientation surface was selected as the hardness spot surface. For any sample block, in the four corners of the selected surface, randomly selected one point in each area for hardness test, and measured the hardness of each sample block for 4 times, and took the average value, as shown in Fig.2. The wear test was carried out with BD4603 wear tester produced by BUCKTOOL COMPANY. The test was carried out in accordance with relevant standards of ISO 7148-2012. The diameter of the sample's wear surface was d=15.5mm. The Korean DEERFSO YA531+ abrasive belt, measuring 100×912 in size with a mesh number of 80, was used. The test was performed at room temperature, with a loading mass of 255g, a rotating speed of 2980r/min, and a wear time of 40 minutes. The microstructure of the metal surface was observed by a double-beam FIB scanning electron microscope (FIB-SEM GX4) manufactured by Thermo Fisher Scientific Shier Technology, USA, and the distribution of interface elements was observed by its own EDS energy spectrum.

Question 2. As far as I can see, there is not a basis for properties for the measured properties. In other words the provided steel by the manufacturer could be further analyzed before any treatment and the effect of any treatment could be also compared to this basis. Alternatively, you could provide measurements from the literature as a basis for comparison.

Response: Thanks for your careful inspection. The Brinell hardness of H13 steel without heat treatment is less than 235HB. Specifically, the Brinell hardness is 229HB, and the Rockwell hardness conversion value is about 20HRC, so the hardness is very low, which is far from that of H13 steel after heat treatment. The impact energy of H13 hot-working die steel with improved ex-factory condition is 30~40J, and its toughness is average. When the steel is heat treated, the impact energy is improved, because the hardness is not comparable, so this mechanical property is not shown in the manuscript. Under the same test conditions, H13 hot-working die steel in the ex-factory condition produced a lot of sparks in less than two minutes during the friction and wear test, and its wear resistance was extremely poor. To sum up, the mechanical properties data of H13 hot-working die steel in the ex-factory condition are too low to be comparable with that of H13 steel after heat treatment, so the relevant original parameters are not shown in the manuscript. We hope that the original mechanical properties of H13 hot-working die steel without any heat treatment can be gradually improved in the future research.

I sincerely appreciate your kind comments very much.

Best Regards

Sincerely Yours

Lipeng Xu

Nov. 3, 2023

Reviewer 3 Report

Comments and Suggestions for Authors

This manuscript presents an experimental work on the determination of the effect of heat treatments on AISI H13 steel. This work is interesting but it is required that various modifications are performed to the manuscript before it can be considered for publication.

Although the Introduction section is generally adequate, the authors are advised to mention various applications of AISI H13 steel, which is also important in manufacturing e.g. for molds. Thus, appropriate references such as: doi.org/10.1016/S0924-0136(02)00861-0, doi.org/10.1016/B978-0-85709-481-0.00006-9, doi.org/10.3390/ma14247863 should be added.

An uppercase "T" is required in the title of Section 2.

The ISO standards used for testing should be mentioned.

In the caption of Figure 3, the authors should mention the element depicted in each photo.

The caption of Figure 4 should be appropriately corrected.

The equation (1) describes the Bragg's, not "Prague" law.

Regarding hardness, appropriate statistical tests should be applied in order to determine the significance of differences between various heat treatments.

Did the authors performed repetitions of measurement of wear and wear rate? Appropriate error bars should be added in Figures 9a and b.

Comments on the Quality of English Language

Moderate corrections are required.

Round 2

Reviewer 2 Report

Comments and Suggestions for Authors

Thank you for your revision.

Reviewer 3 Report

Comments and Suggestions for Authors

The authors have performed most of the necessary modifications to their manuscript, thus it can be recommended for publication.